# Prevalence of Foodborne Bacterial Pathogens and Antibiotic Resistance Genes in Sweets from Local Markets in Iran

**DOI:** 10.3390/foods12193645

**Published:** 2023-10-02

**Authors:** Babak Pakbin, Zahra Amani, Zahra Rahimi, Somayeh Najafi, Behnaz Familsatarian, Alireza Khakpoor, Wolfram Manuel Brück, Thomas B. Brück

**Affiliations:** 1Werner Siemens Chair of Synthetic Biotechnology, Department of Chemistry, Technical University of Munich (TUM), Lichtenberg Str. 4, 85748 Garching bei München, Germany; b.pakbin@tum.de; 2Institute for Life Technologies, University of Applied Sciences Western Switzerland Valais-Wallis, 1950 Sion 2, Switzerland; 3Department of Food Hygiene and Quality of Control, Faculty of Veterinary Medicine, University of Tehran, Tehran 1417614411, Iran; z.amani@ut.ac.ir; 4Department of Food Safety and Health, School of Public Health, Qazvin University of Medical Sciences, Qazvin 34197-59811, Iran; food_supervisor@qums.ac.ir; 5Nutrition and Food Sciences Research Center, Faculty of Pharmacy and Pharmaceutical Sciences, Islamic Azad University, Tehran Medical University (IAUTMU), Tehran 19395-1495, Iran; somayeh_6586@yahoo.com; 6Medical Microbiology Research Center, Qazvin University of Medical Sciences, Qazvin 34197-59811, Iran; behnaz.sattarian@gmail.com; 7Department of Management, Ferdowsi University of Mashhad, Mashhad 9177948974, Iran; a.khakpour56@gmail.com

**Keywords:** foodborne bacterial pathogens, antibiotic resistance genes, sweet products

## Abstract

Background: This study aimed to investigate the prevalences of some important antibiotic-resistance genes (ARGs) and foodborne bacterial pathogens in sweet samples collected from local markets in Iran. Methods: Forty sweet samples were collected. Foodborne pathogens and ARGs were detected in the sweet samples by conventional and multiplex PCR assays using species-specific primers. Results: *Staphylococcus aureus*, *Cronobacter sakazakii*, *Shigella* spp., *Campylobacter jejuni*, and *Campylobacter coli* were detected and identified in 47.5%, 20%, 45%, 5%, and 30% of the sweet samples, respectively. We found *S. aureus* and *Shigella* spp. were the most prevalent bacterial pathogens. *S. aureus* was found to be the most frequent pathogenic bacteria profiled in these samples. We also found a significant correlation between the presence of *C. coli* and *Cr. sakazakii*. We detected the *bla*_SHV_ resistance gene in 97.5% of the sweet samples; however, *bla*_TEM_ was detected in only one sample (2.5%). Conclusions: Regarding these results, we suggest preventive strategies such as implementing automation of food processing; monitoring the personal hygiene and health of food handlers, and testing regularly for antibiotic resistance in raw materials and products.

## 1. Introduction

Foodborne diseases are defined as intestinal or extraintestinal disorders resulting from ingesting or consuming contaminated water, food, or food products [1]. Foodborne illnesses are toxic, infectious, or toxic infectious in nature, and are caused by foodborne pathogens, including different bacterial, fungal, parasitic, and viral species [2]. Consumption of contaminated or raw food products has been associated with several foodborne outbreaks worldwide. The World Health Organization recently reported an estimate that foodborne outbreaks and illnesses cause more than 600 million disease cases and 420,000 deaths annually around the world [3].

Regarding the global number of foodborne illness cases, the major foodborne hazards are *Campylobacter* spp., pathogenic *Escherichia coli* (*E. coli*), non-typhoidal *Salmonella* spp., and *Shigella* spp.; however, *S*. *typhi*, enteropathogenic *E. coli*, enterotoxigenic *E. coli*, *Vibrio cholerae*, and *Campylobacter* spp. are the major hazards considering the global number of deaths caused by bacterial foodborne pathogens. Also, other bacterial agents, such as *Listeria monocytogenes* (*L. monocytogenes*), *Clostridium botulinum*, *Brucella* spp., *Bacillus cereus* (*B. cereus*), *Staphylococcus aureus* (*S. aureus*), and *Clostridium perfringens* (*Cl. perfringens*), have been considered as other major causes of foodborne diseases worldwide [4]. 

On the other hand, resistance to different classes of antibiotics has been regarded as one of the most important challenges in public health and food safety. Foods and foodborne pathogens are the main routes of transmission of ARGs to the human gut microbiota, leading to intestinal drug-resistant infections [5,6]. Beta-lactam resistance genes, including *bla* genes, have been most frequently detected in food samples.

The safety level of foods also depends on the microbial quality of the raw materials used to produce these products. Different foodborne pathogens and antibiotic-resistance genes (ARGs) have been detected in the final products [6,7,8]. Food products such as sweets are primarily prepared and distributed by hand under relatively poor hygienic conditions, and they are susceptible to contamination with foodborne pathogens and ARGs from humans [5,9,10,11].

*S. aureus* strains and toxins are usually isolated from and detected in sweet products since the initial contamination sources of these bacteria are the mucous membranes and skin of humans [12]. 

*S. aureus* has mostly been known as one of the most important food safety challenges and public health concerns, causing intoxication, vomiting, and diarrhea in humans as a foodborne pathogen via the secretion of a heat-stable toxin, alpha-toxin [13]. Other known foodborne bacterial pathogens, such as *Campylobacter* spp., *Shigella* spp., and *Cronobacter sakazakii (Cr. sakazakii*), as a new emerging foodborne pathogen, have been isolated from low-moisture food products such as sweets [14,15]. These foodborne pathogens have been detected in sweet and confectionary products and are also regarded as the main concerns in food safety and public health since they contribute to acute intestinal and extraintestinal diseases in humans. The prevalences of these foodborne bacterial pathogens are significantly and directly linked to the safety levels and hygienic conditions during the production of sweet products [12,13,14,15]. 

Investigation of the prevalences of bacterial pathogens and ARGs in low-moisture food products, especially sweet samples, is strongly limited. On the other hand, there are also limited studies regarding the correlations among the presence of different types of foodborne pathogens in food samples [10]. Therefore, this study aimed to determine the prevalences of and the correlations among the presence of different new emerging foodborne bacterial pathogens, including *C. jejuni*, *C. coli*, *Shigella* spp., *Cr. sakazakii,* and *S. aureus,* and ARGs, in sweet samples.

## 2. Materials and Methods

### 2.1. Sample Collection and Preparation

Forty sweet samples (these traditional Iranian sweets are categorized as solid dry sweets with a moisture content range of 20–26% *w*/*w*), including 4 different subtypes and 3 various brands, were purchased and collected from local markets in Qazvin and Tehran cities, Iran (traditional sweets and confectionaries are mostly produced and consumed in these cities in the center of Iran), between January and April 2022. Samples were immediately transported aseptically in cool boxes containing ice packs to the central microbiology laboratory at the university. 

Each sample was homogenized by using a stomacher BagMixer Lab-blender (Interscience, Saint-Nom-la-Bretèche, France) at 30 °C for 10 min. All samples were diluted 1:10 with phosphate-buffered saline and homogenized again with a stomacher for 5 min at 30 °C [16]. All samples were finally subjected to total DNA extraction and further PCR assays.

### 2.2. Total DNA Extraction

The total DNA in the prepared and diluted sweet samples was extracted by using a SinaClon commercial tissue total DNA extraction kit (SinaClon., Tehran, Iran) according to the manufacturer’s instructions. The instructions of the kit have been optimized for total DNA extraction from food samples. The quantity and quality of the extracted total DNA were assessed by using a NanoDrop spectrophotometer model 1000 (ThermoFisher, Waltham, MA, USA). The optical density ratios of 260/280 of all DNA samples were observed in the range of 1.5–1.8, indicating the high purity of the extracted DNA. The concentrations of all extracted total DNA were adjusted to 50 ng. μL^−1^ by dilution with the addition of sterilized nuclease-free water. The DNA samples were kept at −20 °C for further molecular analysis.

### 2.3. PCR Assays for the Detection and Identification of Different Foodborne Bacterial Pathogens

In the present study, we detected and identified five different foodborne bacterial pathogens, including *C. jejuni*, *C. coli*, *Shigella* spp., *Cr. Sakazakii*, and *S. aureus* by conventional PCR methods (cPCR) using species-specific primers, as shown in Table 1, to directly detect the specific genes in the DNA samples. Identifying *S. aureus* in sweet samples was performed by detecting the *spa* gene in DNA templates according to the method and specific primers previously described and used by Larsen et al. (2008). The *spa* gene encodes Staphylococcal protein A in *S. aureus* strains. Detection of this genetic element proves species-specifically the presence of *S. aureus* in clinical, food, and environmental samples [17]. 

*Cr. sakazakii* was detected in the samples according to the cPCR previously used to detect the *ompA* gene (outer membrane protein virulence factor encoding gene) in powdered infant formula samples by Pakbin et al. (2022) [6]. 

To identify and detect *Shigella* spp. in each sample, the invasion plasmid antigen encoding gene (*ipaH*) was detected by a PCR assay as previously described by Pakbin et al. (2021) [18]. 

*C. coli* and *C. jejuni* strains were identified in sweet samples by detecting the cadF and hipO species-specific genes, as described and employed before by Nafarrate et al. (2021) [19]. 

PCR assays were carried out with 10 µL of PCR 2X master mix (Ampliqon, Odense, Denmark), 2 µL of DNA template (50 ng. μL^−1^), 1 µL of each specific primer (10 mM. μL^−1^), and nuclease-free sterilized water up to the final reaction volume of 20 µL. PCR mixes were subjected to initial denaturation at 95 °C for 5 min, 40 cycles of amplification (denaturation at 95 °C for 1 min, the primer-specific annealing temperature for 30 s, and extension at 72 °C for 1 min), and a final elongation step at 72 °C for 5 min [6,18,19]. The PCR products were characterized by using electrophoresis at 100 v for 1 h on a 1.2% *w/v* agarose gel containing DNA safe-stain (Invitrogen, Carlsbad CA, USA). The PCR products were electrophoresed on agarose gels were evaluated and recorded by using a UV transillumination and gel documentation system (NovinPars Co., Tehran, Iran). Positive and negative samples were used as quality controls for detecting each foodborne pathogen in this study. DNA templates extracted from reference strains, including *S. aureus* (ATCC 25923), *C. jejuni* (ATCC 33291), *C. coli* (ATCC 43478), and *Sh. sonnei* (ATCC 29031), were used as the positive controls, and sterilized distilled water was used as the negative control in this study.

### 2.4. PCR Assays for the Detection of ARGs

Beta-lactamase resistance genes, including *bla*_TEM_, bla_SHV_, *bla*_OXA_, *bla*_CTX-M-1_, *bla*_CTX-M-2_, *bla*_CTX-M-8_, and *bla*_CTX-M-9,_ were detected in sweet samples by using cPCR with thermal cycling programs and specific primers, as shown in Table 1, as previously described by Pakbin et al. (2021) [18]. Reference strains *Klebsiella pneumonia* (ATCC 700603), *Escherichia coli* (ATCC 25922), and *Staphylococcus aureus* (ATCC 25923) harboring all of these ARGs were included and used as quality controls in this study.

### 2.5. Statistical Analysis

Correlations among the different foodborne bacterial pathogen prevalences in the sweet samples were evaluated by using Pearson’s Chi-square test, with a significant difference defined as *p <* 0.05 and one degree of freedom. Fisher`s exact test was used to evaluate significant differences (*p* < 0.05) between the prevalences of different bacterial pathogens, the prevalences of different ARGs, and different bacterial pathogen profiles. All statistical analyses were performed by using SPSS software version 22.0.1 (Chicago, IL, USA). All evaluations and measurements were carried out in triplicate.

## 3. Results

In the present study, we detected some important foodborne pathogens in sweet samples and evaluated the bacterial profiles and correlations between the presence of these pathogens in the collected samples. This study detected *S. aureus* (n = 19), *Cr. sakazakii* (n = 8), *Shigella* spp. (n = 18), *C. jejuni* (n = 2), and *C. coli* (n = 12) in 47.5%, 20.0%, 45.0%, 5.0%, and 30.0% of the sweet samples (N = 40), respectively. Among these bacterial pathogens, *S. aureus* and *Shigella* spp. significantly (*p <* 0.05) showed the highest prevalences compared to the other pathogens. Also, the results in this study illustrated that *C. coli* was significantly (*p <* 0.05) more often detected than *C. jejuni*.

The profiles of the foodborne bacterial pathogens detected in sweet samples in this study are shown in Table 2. A total of 34 (85%) out of 40 sweet samples were contaminated with at least one of the bacterial pathogens. Single bacterial species profiles were detected in 18 samples (45%). Only two foodborne bacterial pathogens were identified in 10 samples (25%). Multiple bacterial profiles, including three or more foodborne pathogens, were detected in 6 (15%) out of the 40 sweet samples. An *S. aureus* single bacterial profile was the significantly (*p <* 0.05) most prevalent profile in this study. A double bacterial profile, including *Shigella* spp. and *C. coli*, was significantly (*p <* 0.05) the most common non-single pathogenic bacterial profile detected in these samples.

We measured the possible correlations between the presence of different pathogens with Pearson’s Chi-square test in this study. Table 3 presents the correlations among the presence of different bacterial pathogens in the sweet samples. Pearson’s correlation analysis showed that the highest significant (*p* < 0.05) positive correlation (0.335) was between the presence of Cr. sakazakii and C. coli in the sweet samples. Significant correlations were not observed between the presence of other bacterial pathogens in this study.

This study detected beta-lactam resistance genes in total DNA extracted from the sweet samples by using cPCR. The *bla*_SHV_ gene was detected in 39 out of 40 sweet samples (97.5%); however, the *bla*_TEM_ gene was only detected in one sample (2.5%). *bla*_OXA_, *bla*_CTX-M-1_, *bla*_CTX-M-2_, *bla*_CTX-M-8_, and *bla*_CTX-M-9_ resistance genes were detected in 25%, 37.5%, 37.5%, and 20% of the sweet samples, respectively.

## 4. Discussion

Foodborne bacterial pathogens are the main causes of intestinal (different types of diarrheal diseases and gastrointestinal disorders) and some extraintestinal illnesses around the world [20]. Regarding the fact that foods and foodborne pathogens are the main routes for the transmission of intestinal pathogens and ARGs to humans and their gut microbiota, the evaluation of the prevalence and presence of different bacterial foodborne pathogens and ARGs in these products is critically needed to implement regularly [21,22]. Considering these facts, we were motivated to investigate the prevalence and incidence of the most important and prevalent foodborne bacterial pathogens and ARGs in sweet products.

The relatively small sample size (N = 40) could be considered this study’s main limitation. We observed the highest *S. aureus* and *Shigella* spp. prevalences in these samples. We detected at least one of these pathogens in 85% of the samples. *S. aureus* was also found to be the most prevalent profile detected in this study. *S. aureus* is an opportunistic pathogen and commensal colonizing the human mucous membranes and skin [23]. Since sweet products are usually prepared and produced traditionally under poor hygienic conditions, contamination of these products with *S. aureus* strains occurs causing gastrointestinal disorders caused by staphylococcal toxins [24]. Houng et al. (2010) detected *S. aureus* in 45 out of 212 ready-to-eat samples (21.2%) collected in Vietnam [25]. Kim et al. (2011) also found that 5.98% of traditional foods produced in Korea were contaminated with *S. aureus* [26]. Mahfoozi et al. (2019) investigated the prevalence of *S. aureus* in some food samples produced in Iran, including meat, dairy, and sweet products, and they detected *S. aureus* in 29.1% of the sweet samples [27]. Hassani et al. (2022) also recently reported contamination with *S. aureus* in 2.67% of the pastry sweets produced in Iran [28]. Using automatic instruments during sweet production and preparation and disinfection of workers’ hands seem to be helpful strategies to reduce the prevalence of *S. aureus* in sweet products [13,23].

*Shigella* is a highly infectious foodborne pathogen belonging to the Enterobacteriaceae family. The main reservoir of this pathogen is the human intestinal system, and it causes mild to severe diarrhea in humans as a facultative intracellular pathogen. *Shigella* species are transmitted to foods via fecally contaminated water or human feces under deplorable hygienic and sanitation conditions [29]. Regarding the low-hygiene preparation conditions (by workers’ hands and usually without any automation) of sweet products, a high prevalence of *Shigella* is predictable [18]. Nisa et al. (2021) also found significant levels of *Shigella* contamination in retail raw food samples in Pakistan. They reported that transmission via raw foods, hospital waste, and unhygienic food handling are the main risk factors for *Shigella* food contamination. This is the first study to report significantly higher levels of *Shigella* contamination in sweet samples; however, several other studies previously reported a high prevalence rate of *S. aureus* contamination in sweet and confectionery samples, as we also observed in the present study. The high prevalence rate of *Shigella* spp. indicates that the health and hygienic practices of personnel and food handlers during the manufacturing process of traditional sweet products should be considered more than other potential risk factors affecting the microbial quality of the products [24,25,26,27,28,29,30,31]. The same strategies can be used to decrease contamination with *Shigella* spp. and *S. aureus* in traditional foods [31]. The *ipaH* gene is also present in entero-invasive *E. coli,* indicating that these strains of *E. coli* might also be detected in this study in addition to *Shigella* spp. [32].

*Cr. sakazakii* is a Gram-negative foodborne pathogen belonging to the Enterobacteriaceae family. It causes severe intestinal and extraintestinal diseases (neonatal meningitis), and is associated with high mortality in infants and neonates [33]. This new emerging foodborne pathogen is relatively resistant to dry environmental conditions and is commonly isolated from dried and powdered foods, such as infant formula, sugar, wheat flour, and dry powdered ingredients [6,11]. In this study, we detected a significant level of contamination with *Cr. sakazakii* in sweet samples. Transmission of this pathogen to foods commonly occurs via contaminated raw food materials and insufficient thermal processing during food preparation [33]. Kim et al. (2011) reported a significant level of contamination (70%) with *Cr. sakazakii* in food samples (Sunshik) produced in Korea [34]. Hassani et al. (2022) also detected *Cr. sakazakii* in 7.14% of pastry sweet samples. Regular microbiological monitoring of raw food materials, especially in terms of detecting *Cr. sakazakii* by rapid methods, is the main strategy for protection against contamination with this foodborne pathogen [28].

*Campylobacter* is another foodborne pathogen associated with enterocolitis in humans. According to the global burden of diarrheal diseases, gastrointestinal disorders caused by this pathogen are very prevalent [35]. Poultry is recognized as the main reservoir of *Campylobacter* species, and poultry products are also known as the main source of contamination with this pathogen in other foods [36]. *C. jejuni* and *C. coli* are the prevalent species of this pathogen, frequently isolated from food products formulated with eggs. Egg whites and yolks are mainly used in the formulation of sweets; therefore, the presence of *Campylobacter* species in sweets is highly probable [37]. In this study, we detected both species of *Campylobacter* in sweet samples; however, the prevalence of *C. coli* was significantly more than that of *C. jejuni*. We also found a high correlation between the presence of *C. coli* and *Cr. sakazakii* in sweet samples. Consequently, the lack of thermal processing can also be considered a risk factor [38]. Systemic monitoring of raw materials, especially the egg whites and yolks used in the formulation of sweet products, in terms of the presence of these pathogens by using rapid assays, can also decrease the risk of contamination with *Campylobacter* species [37,39]. 

To produce sweet products with a high level of microbial quality and safety; automation of food processing, improving the personal hygiene of food handlers, and microbiological monitoring of raw food materials can be considered effective and practical strategies [40]. Notably, automation of sweet production may not be possible due to the fact that they are produced in small workshops using traditional methods; therefore, it should be suitable to suggest industrializing the production processes of confectionaries and sweets to improve the safety levels of these products. On the other hand, using uncontaminated raw materials such as flour, egg whites, spice powders, and dried fruits and nuts could significantly affect the hygienic production conditions of sweet products. Also, the correlations among the presence of different foodborne pathogens indicate that the same hygienic strategies can be considered to provide for the safety of these products. 

Other foodborne pathogens, such as *Cl. perfringens, B. cereus*, and *L. monocytogenes*, and also some non-bacterial (fungal and viral) foodborne pathogens, have also been detected and identified in low-moisture sweet and confectionary products at relatively lower prevalence levels. However, bacterial pathogens are still considered the leading human health-threatening concern in these products [4,15,37,41].

In this study, we detected beta-lactam resistance genes, including *bla*_TEM_, *bla*_SHV_, *bla*_OXA_, *bla*_CTX-M-1_, *bla*_CTX-M-2_, *bla*_CTX-M-8_, and *bla*_CTX-M-9_ in sweet samples. Beta-lactamases are the main enzymes responsible for the resistance of Gram-negative bacteria, such as *Cronobacter*, *Campylobacter*, and *Shigella* spp., against the beta-lactam antibiotics [42]. Pakbin et al. (2021) also detected the *bla*_SHV_ gene as the most prevalent ARG in low-moisture food samples. The *bla*_SHV_ gene, which was the dominant beta-lactam resistance gene in the sweet samples in this study, encodes resistance against amoxicillin and amoxicillin-clavulanic antibiotics [18]. There are limited studies concerning the presence of different ARGs in sweet samples; however, several studies reported and investigated the presence of ARGs in food samples such as processed dairy products. Wu et al. (2020) investigated the prevalences of ARGs in high-moisture sweet samples, and they found the largest prevalence of the *tetW*, *tetT,* and *tetA46* genes in these samples are associated with resistance to antibiotics belonging to the class of tetracyclines. They also detected the *erm41* and *tlrC* genes associated with resistance against the antibiotic class of macrolides [42]. Regarding the small sample size as the main limitation of this study, comprehensive epidemiological studies are suggested to be implemented in the future to investigate the prevalences of different foodborne bacterial, fungal, and viral pathogens in sweet products.

## 5. Conclusions

We found that sweet samples were significantly more contaminated with *S. aureus* and *Shigella* spp. *S. aureus* was the most frequent pathogenic bacteria in these profiles. We also found a significant correlation between the presence of *C. coli* and *Cr. sakazakii* in sweet samples. The *bla*_SHV_ gene was detected most frequently in sweet samples; adversely, the *bla*_TEM_ gene was only detected in one sample. Considering these results, we suggest automation of food processing, improving the personal hygiene of workers, evaluation of microbial contamination of raw materials, and regular monitoring of ARGs in raw materials and the final products to produce safe sweet products with higher levels of microbial quality. This is the first study to report a high prevalence rate of *Shigella* spp. in traditional sweet products; consequently, preventive strategies regarding the health and hygiene of the manufacturing personnel are suggested to be more considered than other strategies during the production of traditional sweets. Comprehensive epidemiological studies concerning the prevalences of different foodborne bacterial, fungal, and viral pathogens in various sweet types are also highly recommended to be implemented in the future.

## Figures and Tables

**Table 1 foods-12-03645-t001:** Species-specific primers used to detect foodborne pathogens and beta-lactam resistance genes in sweet samples.

Gene	Primer	Sequence (5′ → 3′)	Reference
*spa*	*spa1*	TAAAGACGATCCTTCGGTGAGC	[17]
	*spa2*	CAGCAGTAGTGCCGTTTGCTT	
*ompA*	*ompAF*	GGATTTAACCGTGAACTTTTCC	[6]
	*ompAR*	CGCCAGCGATGTTAGAAGA	
*ipaH*	*ipaHF*	GTTCCTTGACCGCCTTTCCGATACCGTC	[18]
	*ipaHR*	GCCGGTCAGCCACCCTCTGAGAGTAC	
*hipO*	*hipOF*	AATGCACAAATTTGCCTTATAAAAGC	[19]
	*hipOR*	TNCCATTAAAATTCTGACTTGCTAAATA	
*cadF*	*cadFF*	GAGAAATTTTATTTTTATGGTTTAGCTGGT	[19]
	*cadFR*	ACCTGCTCCATAATGGCCAA	
*bla* _TEM_	blaTEMF	CATTTCCGTGTCGCCCTTATTC	[18]
	blaTEMR	CGTTCATCCATAGTTGCCTGAC	
*bla* _SHV_	blaSHVF	AGCCGCTTGAGCAAATTAAAC	[18]
	blaSHVR	ATCCCGCAGATAAATCACCAC	
*bla* _OXA_	blaOXAF	GGCACCAGATTCAACTTTCAAG	[18]
	blaOXAR	GACCCCAAGTTTCCTGTAAGTG	
*bla* _CTX-M-1_	CTXM1F	TTAGGAARTGTGCCGCTGYA	[18]
	CTXM1R	CGATATCGTTGGTGGTRCCAT	
*bla* _CTX-M-2_	CTXM2F	CGTTAACGGCACGATGAC	[18]
	CTXM2R	CGATATCGTTGGTGGTRCCAT	
*bla* _CTX-M-8_	CTXM8F	AACRCRCAGACGCTCTAC	[18]
	CTXM8R	TCGAGCCGGAASGTGTYAT	
*bla* _CTX-M-9_	CTXM9F	TCAAGCCTGCCGATCTGGT	[18]
	CTXM9R	TGATTCTCGCCGCTGAAG	

**Table 2 foods-12-03645-t002:** Profiles of different foodborne bacterial pathogens detected in sweet samples.

	Foodborne Bacterial Pathogen Profile ^a^	Positive Samples (%) (n)
Single	SA	25 (10)
	SH	15 (6)
	CR	2.5 (1)
	CC	2.5 (1)
Double	SH + CC	10 (4)
	SA + SH	5 (2)
	SA + CC	2.5 (1)
	SA + CJ	2.5 (1)
	CR + CC	2.5 (1)
	SH + CR	2.5 (1)
Multiple	SA + CR + SH	2.5 (1)
	CR + SH + CC	2.5 (1)
	SA + SH + CC	2.5 (1)
	SA + CR + SH + CC	5 (2)
	SA + CR + CJ + CC	2.5 (1)

^a^ S. aureus, SA; Shigella spp., SH; Cr. sakazakii, CR; C. jejuni, CJ; C. coli, CC.

**Table 3 foods-12-03645-t003:** Correlations among the presence of different foodborne bacterial pathogens in the sweet samples.

	*C. coli*	*C. jejuni*	*Cr. sakazakii*	*Shigella spp.*	*S. aureus*
*C. coli*	1.000	0.100	0.355 *	0.285	−0.076
*C. jejuni*		1.000	0.172	−0.208	0.241
*cr. sakazakii*			1.000	0.176	0.025
*Shigella spp.*				1.000	−0.257
*S. aureus*					1.000

* *p <* 0.05.

## Data Availability

We confirm that all data and findings of this study are available within the article.

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
