# Peer review of "Prevalence of Foodborne Bacterial Pathogens and Antibiotic Resistance Genes in Sweets from Local Markets in Iran"

_foods, 2023, doi:10.3390/foods12193645_

Round 1

Reviewer 1 Report

Comments and Suggestions for Authors

The manuscript presents a study to investigate the prevalence of some foodborne pathogens and antibiotic-resistance genes in sweets. The significance and novelty need to be further clarified. There are also some issues that need to be addressed.

Line 71, the abbreviation "S. aureus" should be given after the first time of the full name in line 54. The same for the following bacterial names.

Line 105, Why were these five strains selected in the study?

Line 194, how the five pathogens in the study were determined needs citations. Are there any previous publications that showed the five pathogens are the most important?

The significance of the study on the prevalence of the five pathogens needs to be further clarified. Is there really a safety concern or will it cause foodborne outbreaks when those pathogens are present in the sweets? Since there is a high prevalence of those pathogens in the sweets, does it mean the safety of the selected 40 samples could be a concern? Do the samples represent the common situation of the sweets on the market?

For the 40 samples, were they from the same brand? Were they the same type or independently different? Needs more description on the samples. Since the sample size is relatively small (only 40), how can the samples represent the common situation of the sweets?

The significance and novelty of the study should be further discussed. The conclusion of the study is to recommend proper handling and hygiene for sweet processors, but they are common practices. What's the significance of the findings?

Comments on the Quality of English Language

Overall it is well-written in terms of the quality of the English language.

Author Response

Dear Reviewer 1,

Thank you very much for your valuable and practical comments. The manuscript has been revised considering all of your revisions and points of view. All revisions have been addressed and can be found in the revised track change format main text file. Point by point responses to each revision have been described below:

The significance and novelty need to be further clarified. There are also some issues that need to be addressed.

Investigation of the prevalence of bacterial pathogens and ARGs in low-moisture food products especially sweet samples is strongly limited. On the other hand, there is also limited studies regarding the correlation among the presence of different foodborne pathogens in food samples. This novelty and significance of the study is mentioned and clarified in the introduction section.

Line 71, the abbreviation "S. aureus" should be given after the first time of the full name in line 54. The same for the following bacterial names.

The abbreviation has been corrected accordingly.

Line 105, Why were these five strains selected in the study?

These five strains have been selected because these foodborne pathogens were detected in sweet and confectionary products and are also regarded as the main concerns in food safety and public health since they contribute to acute intestinal and extra-intestinal diseases in humans. This explanation has been added into the manuscript, the introduction section.

Line 194, how the five pathogens in the study were determined needs citations. Are there any previous publications that showed the five pathogens are the most important?

Separately, there are publications that showed the presence of these pathogens in sweet and confectionary products and these publications have been cited into the manuscript. This is the first study that investigate all of these pathogens simultaneously in sweet samples. This explanation has been added into the last paragraph of the introduction section.

The significance of the study on the prevalence of the five pathogens needs to be further clarified. Is there really a safety concern or will it cause foodborne outbreaks when those pathogens are present in the sweets? Since there is a high prevalence of those pathogens in the sweets, does it mean the safety of the selected 40 samples could be a concern? Do the samples represent the common situation of the sweets on the market?

For the 40 samples, were they from the same brand? Were they the same type or independently different? Needs more description on the samples. Since the sample size is relatively small (only 40), how can the samples represent the common situation of the sweets?

Investigation of the prevalence of bacterial pathogens and ARGs and the correlation among the presence of different foodborne pathogens in low-moisture food products especially sweet samples are strongly limited. The relatively small sample size (N=40) could be considered as the main limitation of this study this study's main limitation. Regarding the small sample size as the main limitation of this study, we also suggested that comprehensive epidemiological studies to be implemented in the future to investigate the prevalence of different foodborne bacterial, fungal and viral pathogens in sweet products. These explanations have been added into the discussion section of the manuscript.

The significance and novelty of the study should be further discussed. The conclusion of the study is to recommend proper handling and hygiene for sweet processors, but they are common practices. What's the significance of the findings?

The significance and novelty of this study and our findings have been addressed in introduction, discussion and conclusion sections of the manuscript. Also, more strategies against the bacterial contamination during the sweet production have been suggested and discussed in the manuscript.

Reviewer 2 Report

Comments and Suggestions for Authors

The article titled " Prevalence of foodborne bacterial pathogens and antibiotic resistance genes in sweets" While the article seems well-structured and contains a wealth of information, there are areas where it could benefit from improvements:

Plagiarism:  24%

Abstract:

  • Title Integration: Link the title to the abstract by specifying that the research focused on sweets from local markets in Iran.
  • Clarity: The abstract could be more concise. Trim out unnecessary details and emphasize the main findings and implications.
  • Terminology: Ensure consistent use of terms like 'antibiotic-resistance genes'.

Introduction:

  • Consolidation of Information: Some information is repetitive. For example, the discussion on the role of food and foodborne pathogens in transmitting antibiotic resistance could be mentioned once and elaborated upon to avoid redundancy.
  • Flow and Coherence: The introduction could benefit from a more linear progression of information. Start with the broader context of foodborne illnesses and pathogens, then narrow down to antibiotic resistance, and finally zone in on the importance of studying sweets.
  • Updated References: Make sure that the most recent and relevant studies are cited. Using outdated references could reduce the perceived novelty of the research.
  • Relevance: Focus on making every piece of information directly relevant to the main aim of the research. For instance, discussing the importance of hygiene in sweet production could be linked directly to the prevalence of bacterial pathogens in sweets.

Methodology:

  • Clarity and Precision: Simplify complex methods to make them easier to understand for a wider audience. For instance, the description of PCR assays could be streamlined for better readability.
  • Detailing: While detailed methods are crucial, avoid overly technical language without providing explanations. Consider using supplementary materials for very detailed procedures.
  • Organization: Consider using subheadings for various stages of the methodology, like "Sample Collection", "DNA Extraction", "PCR Assays", etc. This will make the methodology easier to follow.
  • Consistency in Terminology: Ensure that terms used in the methods section are consistent throughout. For instance, always refer to 'PCR assays' or 'conventional PCR method' in a consistent manner.
  • Statistical Analysis: Specify the criteria and rationale for using specific statistical tests. Ensure that the methodology can be replicated based on the provided details.
  • Quality Control: The mention of quality controls is excellent. However, provide more details or references about how these controls were determined and their relevance.
  • Sample Information: Elaborate on why particular cities in Iran were chosen for sample collection. Provide details like the season, specific types of sweets, etc., if relevant.

Results:

  • Clarity: Ensure that the results are presented in a systematic, straightforward manner.
  • Table referencing: Reference tables properly within the text. For instance, "described in Table 2" should be "as shown in Table 2."
  • Formatting: Use appropriate symbols for percentages, e.g., "97.5%" instead of "97.5 %."
  • Consistency: Maintain a consistent naming convention for bacterial strains and antibiotic-resistance genes.

Discussion:

·        Depth of Analysis: Delve deeper into the implications of the findings rather than just stating them.

·        Comparison with Previous Work: More comparisons with existing literature can strengthen the discussion. Only a few existing studies are referenced, and greater emphasis on how the current research compares or contrasts with prior work would enhance the discussion.

·        Causative Factors: Investigate potential reasons for the prevalence of certain pathogens over others in the sweets.

·        Clarify limitations: Highlight limitations more explicitly. For instance, the mention of the small sample size should be more central to the discussion.

·        Recommendations: Along with the strategies mentioned, provide more in-depth recommendations for avoiding contamination, especially in the context of the study's findings.

Conclusions:

·        Summarization: The conclusion should succinctly summarize the main findings without introducing new data.

·        Implications: Emphasize the broader implications of the research findings for public health and the sweets industry.

·        Future Recommendations: Offer suggestions for further studies, potential mitigation strategies, or areas that would benefit from additional research.

·        General clarity: Streamline the conclusion to provide a clear takeaway for the reader without getting too detailed or repetitive.

General:

·        Language & Grammar: Some sentences are long and convoluted. Simplifying and breaking them down would make the content more accessible.

·        Citation Consistency: Ensure that all studies referenced in the discussion have corresponding citations and are included in the reference list.

·        Terminology: Ensure consistent use of technical terms, abbreviations, and acronyms. Introduce them properly the first time they appear in the text.

·        Detailed tables: The tables (like Table 2 and Table 3) should be presented with clear headings, labels, and footnotes if necessary. Ensure that any abbreviations used in tables are defined either within the table or in a footnote.

These improvements aim to enhance the clarity, depth, and scientific rigor of the article.

Comments on the Quality of English Language

Moderate editing of English language required

Author Response

Dear Reviewer 2,

Thank you very much for your valuable and useful comments. The manuscript has been revised considering all of your revisions and points of view. All revisions have been addressed and can be found in the revised track change format main text file. Point by point responses to each revision have been described below:

Abstract:

Title Integration: Link the title to the abstract by specifying that the research focused on sweets from local markets in Iran.

It has been revised accordingly in the text.

Clarity: The abstract could be more concise. Trim out unnecessary details and emphasize the main findings and implications.

It has been revised accordingly in the text.

Terminology: Ensure consistent use of terms like 'antibiotic-resistance genes'.

It has been revised accordingly in the text.

Introduction:

Consolidation of Information: Some information is repetitive. For example, the discussion on the role of food and foodborne pathogens in transmitting antibiotic resistance could be mentioned once and elaborated upon to avoid redundancy.

It has been revised accordingly in the text.

Flow and Coherence: The introduction could benefit from a more linear progression of information. Start with the broader context of foodborne illnesses and pathogens, then narrow down to antibiotic resistance, and finally zone in on the importance of studying sweets.

It has been revised accordingly in the text.

Updated References: Make sure that the most recent and relevant studies are cited. Using outdated references could reduce the perceived novelty of the research.

The references have been updated and revised accordingly in the text.

Relevance: Focus on making every piece of information directly relevant to the main aim of the research. For instance, discussing the importance of hygiene in sweet production could be linked directly to the prevalence of bacterial pathogens in sweets.

It has been revised accordingly in the text.

Methodology:

Clarity and Precision: Simplify complex methods to make them easier to understand for a wider audience. For instance, the description of PCR assays could be streamlined for better readability.

It has been revised accordingly in the text and more information are added.

Detailing: While detailed methods are crucial, avoid overly technical language without providing explanations. Consider using supplementary materials for very detailed procedures.

It has been revised accordingly in the text and more information are added.

Organization: Consider using subheadings for various stages of the methodology, like "Sample Collection", "DNA Extraction", "PCR Assays", etc. This will make the methodology easier to follow.

It has been revised accordingly in the text.

Consistency in Terminology: Ensure that terms used in the methods section are consistent throughout. For instance, always refer to 'PCR assays' or 'conventional PCR method' in a consistent manner.

It has been revised accordingly in the text.

Statistical Analysis: Specify the criteria and rationale for using specific statistical tests. Ensure that the methodology can be replicated based on the provided details.

It has been revised accordingly in the text.

Quality Control: The mention of quality controls is excellent. However, provide more details or references about how these controls were determined and their relevance.

All reference strains which were used in this study have been added into the manuscript in the methods section and revised accordingly in the text.

Sample Information: Elaborate on why particular cities in Iran were chosen for sample collection. Provide details like the season, specific types of sweets, etc., if relevant.

It has been revised accordingly in the text and more information are added.

Results:

Clarity: Ensure that the results are presented in a systematic, straightforward manner.

It has been revised accordingly in the text.

Table referencing: Reference tables properly within the text. For instance, "described in Table 2" should be "as shown in Table 2."

It has been revised accordingly in the text.

Formatting: Use appropriate symbols for percentages, e.g., "97.5%" instead of "97.5 %."

It has been revised accordingly in the text.

Consistency: Maintain a consistent naming convention for bacterial strains and antibiotic-resistance genes.

It has been revised accordingly in the text.

Discussion:

  • Depth of Analysis: Delve deeper into the implications of the findings rather than just stating them.

It has been revised accordingly in the text and more information are added.

  • Comparison with Previous Work: More comparisons with existing literature can strengthen the discussion. Only a few existing studies are referenced, and greater emphasis on how the current research compares or contrasts with prior work would enhance the discussion.

It has been revised accordingly in the text and more information are added.

  • Causative Factors: Investigate potential reasons for the prevalence of certain pathogens over others in the sweets.

It has been revised accordingly in the text and more information are added.

  • Clarify limitations: Highlight limitations more explicitly. For instance, the mention of the small sample size should be more central to the discussion.

Investigation of the prevalence of bacterial pathogens and ARGs and the correlation among the presence of different foodborne pathogens in low-moisture food products especially sweet samples are strongly limited. The relatively small sample size (N=40) could be considered as the main limitation of this study this study's main limitation. Regarding the small sample size as the main limitation of this study, we also suggested that comprehensive epidemiological studies to be implemented in the future to investigate the prevalence of different foodborne bacterial, fungal and viral pathogens in sweet products. These explanations have been added into the discussion section of the manuscript.

  • Recommendations: Along with the strategies mentioned, provide more in-depth recommendations for avoiding contamination, especially in the context of the study's findings.

It has been revised accordingly in the text and more information are added.

Conclusions:

  • Summarization: The conclusion should succinctly summarize the main findings without introducing new data.

It has been revised accordingly in the text.

  • Implications: Emphasize the broader implications of the research findings for public health and the sweets industry.

It has been revised accordingly in the text and more information are added.

  • Future Recommendations: Offer suggestions for further studies, potential mitigation strategies, or areas that would benefit from additional research.

It has been revised accordingly in the text and more information are added.

  • General clarity: Streamline the conclusion to provide a clear takeaway for the reader without getting too detailed or repetitive.

It has been revised accordingly in the text.

General:

  • Language & Grammar: Some sentences are long and convoluted. Simplifying and breaking them down would make the content more accessible.

Language and grammar of the text have been checked and revised throughout the manuscript.  

  • Citation Consistency: Ensure that all studies referenced in the discussion have corresponding citations and are included in the reference list.

It has been checked and revised accordingly in the text.

  • Terminology: Ensure consistent use of technical terms, abbreviations, and acronyms. Introduce them properly the first time they appear in the text.

It has been checked and revised accordingly in the text.

  • Detailed tables: The tables (like Table 2 and Table 3) should be presented with clear headings, labels, and footnotes if necessary. Ensure that any abbreviations used in tables are defined either within the table or in a footnote.

It has been checked and revised accordingly in the text.

These improvements aim to enhance the clarity, depth, and scientific rigor of the article.

Round 2

Reviewer 1 Report

Comments and Suggestions for Authors

I thank the authors for revising the manuscript and addressing those concerns. There are some questions that need to be further explained. 

For the sweet samples, what type of sweets are they? (solid candy, soft candy, wrapped individually, etc)? The different types of sweets might affect the pathogen profile of the sample. Need more information about the samples.

The presented study found sweet samples were significantly contaminated with S. aureus and Shigella. How was this result compared with previous studies on sweet samples?

The suggestions from the study are automated food preparation plans, personal hygiene, microbial evaluation of raw materials, and regular monitoring which are actually common practices in the food industry. What are the further or more significant impacts of the results?

Comments on the Quality of English Language

Just need some minor edits.

Author Response

Dear Reviewer 1,

Thank you very much for your valuable comments. The manuscript has been revised according to your comments and all revisions have been addressed and implemented as described below:

  • The type of sweet samples were solid dry sweets with a moisture content range of 20-26% w/w. This statement has been added to the manuscript.
  • The results have been compared with other previous studies regarding the prevalence of Shigella spp. and S. aureus in sweet samples. However, this is the first study that reports significantly higher levels of Shigella contamination in sweet samples. This result has also been compared to the other previous studies. The discussion and conclusions sections have been revised accordingly. 
  • The high prevalence rate of Shigella spp. indicates that the health and hygienic practices of personnel and food handlers during the manufacturing process of traditional sweet products should be considered more than other potential risk factors affecting the microbial quality of the products. This statement has been added to the discussion and conclusions sections of the manuscript as one of the significant impacts of this study. 
